Predictability and transferability of local biodiversity environment relationships

http://orcid.org/0000-0002-7569-1390 Jung Martin jung@iiasa.ac.at
Biodiversity and Natural Resources Program, International Institute for Applied Systems Analysis (IIASA) , Laxenburg, Lower Austria , Austria
Ward Eric
Electronic publication date: 2022 Aug 23
Publication date: 2022
Volume: 10
Electronic Location ID: e13872
Received 2022 May 4; Accepted 2022 Jul 19
Copyright: © 2022 Jung
Copyright year: 2022
Copyright holder: Jung
License: This is an open access article distributed under the terms of the Creative Commons Attribution License, which permits unrestricted use, distribution, reproduction and adaptation in any medium and for any purpose provided that it is properly attributed. For attribution, the original author(s), title, publication source (PeerJ) and either DOI or URL of the article must be cited.
License URL: https://creativecommons.org/licenses/by/4.0/

Keywords: Spectral-diversity, Biodiversity-productivity, Transferability, Remote-sensing, PREDICTS, Extrapolation, Biodiversity indicators, Prediction uncertainty

Funding: The author received no funding for this work.

==============================
Background

Biodiversity varies in space and time, and often in response to environmental heterogeneity. Indicators in the form of local biodiversity measures–such as species richness or abundance–are common tools to capture this variation. The rise of readily available remote sensing data has enabled the characterization of environmental heterogeneity in a globally robust and replicable manner. Based on the assumption that differences in biodiversity measures are generally related to differences in environmental heterogeneity, these data have enabled projections and extrapolations of biodiversity in space and time. However so far little work has been done on quantitatively evaluating if and how accurately local biodiversity measures can be predicted.

Methods

Here I combine estimates of biodiversity measures from terrestrial local biodiversity surveys with remotely-sensed data on environmental heterogeneity globally. I then determine through a cross-validation framework how accurately local biodiversity measures can be predicted within (“predictability”) and across similar (“transferability”) biodiversity surveys.

Results

I found that prediction errors can be substantial, with error magnitudes varying between different biodiversity measures, taxonomic groups, sampling techniques and types of environmental heterogeneity characterizations. And although errors associated with model predictability were in many cases relatively low, these results question–particular for transferability–our capability to accurately predict and project local biodiversity measures based on environmental heterogeneity. I make the case that future predictions should be evaluated based on their accuracy and inherent uncertainty, and ecological theories be tested against whether we are able to make accurate predictions from local biodiversity data.

Introduction

Local biodiversity on land is known to vary with environmental heterogeneity (Hillebrand, 2004; Stein & Kreft, 2015; Holt et al., 2017), often quantified as difference in availability and variability of resources available to a species. These resources, such as nutrients, water, energy characterize the suitable habitat where the population of a species can persist and commonly include the key availability and diversity of components of ecosystem functioning (Stein & Kreft, 2015; Regos et al., 2022). Several theories have been postulated as possible source of the relationship of environmental heterogeneity with local biodiversity. These include, among others, the widely tested species-energy (Hurlbert, 2004; Evans, Warren & Gaston, 2005; Duncan et al., 2015), the species spectral-heterogeneity (Oldeland et al., 2010; Rocchini et al., 2010) or the species-geodiversity hypotheses (Alahuhta, Toivanen & Hjort, 2020). The assumption is that habitats with greater heterogeneity provide more niches for diverse sets of species and species population expansions. However, despite a number of global meta-analyses on the relationship between environmental heterogeneity and local biodiversity for plant, bird and mammal species (Stein, Gerstner & Kreft, 2014; Duncan et al., 2015), it has rarely been comprehensively investigated how predictable and transferable these relationships are, especially across taxonomic groups and different biodiversity measures more generally.

Predictions made by statistical models are key for our understanding of the living world and for the creation of outputs relevant for conservation management (Miller et al., 2004; Houlahan et al., 2017). Because of the evermore increasing demand for scenarios and spatial maps by policy makers and land managers, biodiversity modellers often need to rely on inter- and extrapolations of model predictions across space and time (Miller et al., 2004). These predictions need to be precise and accurate enough for the context and decisions they are meant to inform (Santini et al., 2021). Thus model predictions should be investigated for their predictability, e.g., a model’s ability to accurately predict correlative relationships within the same spatial and/or temporal context by withholding some parts of the data (as in cross-validation procedures), and transferability, e.g., the capacity to produce accurate predictions for conditions dissimilar to those of the data for which a model was trained by withholding data outside the range used for calibration or making use of independently collected data (Petchey et al., 2015; Jung et al., 2017; Yates et al., 2018; Tredennick et al., 2021). And yet, model predictability and transferability is rarely consistently assessed and, when studied in more detail, results rarely look promising.

There is increasing evidence that models using variables of environmental heterogeneity, in the context of this work defined as characterizations of local habitats by vegetation and land-surface conditions, often fail to accurately predict and transfer local biodiversity measures. Studies have found that the predictability of local biodiversity as function of a difference in environmental heterogeneity are highly variable between geographic regions (Phillips, Newbold & Purvis, 2017) and local contexts (Duncan et al., 2015; Jung et al., 2017). Similarly, transferability of model predictions to spatial or temporally distinct regions has long been recognized as key issue for species distribution models (Zurell, Elith & Schröder, 2012; Mesgaran, Cousens & Webber, 2014; Regos et al., 2019) or models using local and regional biodiversity measures (Parmentier et al., 2011; Schmidtlein & Fassnacht, 2017). Despite the development of techniques for assessing the novel parameter space of a model (Zurell, Elith & Schröder, 2012; Meyer & Pebesma, 2021), the limited uptake of modellers to evaluate and present model uncertainty can hinder the application and affect trust in biodiversity model predictions (Rapacciuolo, 2019). Here, one key reason and shortcoming in doing so in macroecological studies are the various and context-specific ways in which environmental heterogeneity is quantified (Stein & Kreft, 2015), preventing assessments of predictability and transferability.

Recent advances in remote sensing and cloud-processing have enabled the robust quantification of environmental heterogeneity at high spatial and temporal resolution (Gorelick et al., 2017; Randin et al., 2020; Regos et al., 2022). Through repeated satellite observations, measures of environmental heterogeneity, such as differences in photosynthetic activity or spectral variability as proxies for vegetation cover, vegetation condition and structure and overall variability of land surfaces (Rocchini et al., 2010; Radeloff et al., 2019), can be robustly quantified. On their own they can be considered continuous representations of contrasts in land cover and land use (Hansen et al., 2000; Jung, Scharlemann & Rowhani, 2020) and ecosystem functioning (Regos et al., 2022), while also being related to key species population processes (Pettorelli et al., 2005). Although such remotely-sensed measures do not cover all types of environmental heterogeneity (Stein, Gerstner & Kreft, 2014), they can be exogenously quantified in space and time and have often been incorporated in statistical models for the prediction of species distributions (Cord et al., 2013; He et al., 2015; Regos et al., 2022) or to infer differences in local biodiversity measures (Oldeland et al., 2010; Goetz et al., 2014; Rocchini, Hernández-Stefanoni & He, 2015; Jung et al., 2019; Jung, Scharlemann & Rowhani, 2020). Remote sensing measures of environmental heterogeneity can therefore–opposed to study-specific predictors commonly included in ecological meta-analysis–serve as a globally consistent predictor for evaluating biodiversity environment relationships (Duncan et al., 2015). With the availability of new global databases on local biodiversity in-situ observations (Hudson et al., 2017), it has become possible to comparably investigate the predictability and transferability of biodiversity environment relationships across taxonomic groups and biodiversity measures.

There are a number of shortcomings in previous analyses on the predictability and transferability of local biodiversity environment relationships. Most studies have (a) focussed on effect sizes among studies (e.g., strength of inference), rather than the predictability and transferability of this relationships (Tredennick et al., 2021), (b) tended to focus mostly on species richness (Stein, Gerstner & Kreft, 2014), thus ignoring other biodiversity measures such as abundance or differences in species assemblage composition, (c) used variables of varying origin to capture effects of changes in environmental heterogeneity on biodiversity (Supp & Ernest, 2014; Shackelford et al., 2017) or have (d) focussed only on regional extents and single taxonomic groups such as birds, butterflies or plants (Kerr, Southwood & Cihlar, 2001; Oldeland et al., 2010; Goetz et al., 2014; Schmidtlein & Fassnacht, 2017). Quantitatively addressing these issues is key, if we are to understand in which cases spatial and/or temporal predictions of local biodiversity measures are reliable and accurate.

In this study I investigate the predictability and transferability of model-based predictions on local biodiversity environment relationships. The expectation is that (i) predictability is generally stronger than transferability, (ii) transferability of species-environment relationships affects some biodiversity measures and taxonomic groups more than others, and that (iii) unexplained variation in the tested relationships is predominantly linked to differences in study design, e.g., spatial scale and sampling duration. To test this, I combine local biodiversity data from globally distributed surveys with remotely-sensed environmental predictors quantifying photosynthetic activity (Evans, Warren & Gaston, 2005; Stein, Gerstner & Kreft, 2014; Duncan et al., 2015) and spectral variability (Rocchini et al., 2010); predictors that represent a continuous characterization of the availability of resources and land surface modifications. Using variations of generalized linear and additive models, I assess the predictability, quantified as overall and within-study reduction in prediction error, and transferability, quantified as reduction in prediction error between different studies of comparable design but identical taxonomic groups (Fig. 1). The aim of this work is thus to comparatively assess the strength and generality of local biodiversity-environment relationships at a global scale, which hopefully stimulates a debate on whether predicted local biodiversity measures, such as total site-based abundance or richness, can accurately be predicted or transferred to unsampled regions.

Figure 1 Schematic of the analysis framework showing the distribution of two hypothetical studies and their sites at which a biodiversity measure and environmental predictor has been calculated.

(A) Hypothetical studies are coloured in orange and red and the Normalized Difference Vegetation Index (NDVI) is shown as example of a remotely sensed environmental predictor. A simplified procedure for investigating the (B) predictability and (C) transferability of local biodiversity-environment relationships is shown. For (B) ‘testing’ sites within a studies are removed at random, regressions refitted and the within-study prediction error quantified in relation to study properties. In contrast, in (C) regression fits from one study (orange) are used to predict permuted biodiversity estimates in another study (red) that have been removed (beige), with the prediction error quantified in relation to study properties (i.e., scale, sampling length, indicated by different icons).

Materials and Methods

Biodiversity data preparation

For data on biodiversity I took species assemblage data from the global Projecting Responses of Ecological Diversity In Changing Terrestrial Systems (PREDICTS) database (Hudson et al., 2017), which contains records of species occurrence and abundance at spatial-explicit sites ‘sites’ as reported in published ‘studies’. PREDICTS includes only studies which differ in ‘land-use’ and/or ‘land-use intensity’ and have spatial and temporal information associated with them, e.g., sampling extent and date of sampling (Hudson et al., 2014). Studies in the PREDICTS database vary widely in study properties, notably in taxonomic coverage (studies contain data on terrestrial species of invertebrates, plants, birds, mammals, reptiles and amphibians), spatial grain (0.05–39,150 m, median = 70 m ± 97 MAD), sampling start (1984–2013), sampling effort (>0–4,382 days, median = 91 days) and methodology (flight traps, transects,…). Owing to these differences, a hierarchical modelling framework is usually necessary when analysing biodiversity estimates from databases such as PREDICTS (Purvis et al., 2018).

For each study j and site i in the PREDICTS database, I calculated four different site-based measures of local biodiversity: total Species richness ( Si), total log-transformed abundance ( log10Ai), the arcsine square root transformed probability of interspecific encounter as measure of assemblage evenness ( sin−1PIEi) and the logit transformed pairwise Sørensen similarity index as measure of difference in assemblage composition ( logitSIMi−in). Similar to previous studies I assumed that, in the few cases where within-study study effort differs among sites, the abundance of species individuals increases linearly with sampling effort (Newbold et al., 2015). In cases where the sampling extent of a site is missing in the PREDICTS database, I approximated the mean sampling extent using a heuristic that fills missing estimates with the average used within studies of the same sampling method and/or taxonomic group. Earlier work has shown that this approximation can accurately fill missing sampling extents (Jung et al., 2019). Lastly, I created, based on the taxonomic group and sampling method attributed to a study in the PREDICTS database, a new factor variable that groups studies of comparable method, unit and broad taxonomic grouping (Table S1), such as for instance studies involving bird individuals that were counted using point counts. I realize that not all differences in sampling techniques can be attributed to this new contrast between sites and therefore post-hoc analyse the contribution of differing sampling methods in explaining the cross-validated model error (see statistical analysis). In total I used data from 564 different studies and 25,849 sites, with the median number of sites per study being 18 (IQR = 35, see also Fig. S6).

Environmental predictors

In this work I exclusively used remotely-sensed environmental predictors, namely photosynthetic activity and spectral variability, which are (1) available at adequate spatial resolution, (2) consistently quantified at global extent in comparable units, (3) temporally explicit, often differing between years, (4) correlate with differences in local biodiversity (Duncan et al., 2015; Jung et al., 2019) and land use (Mueller et al., 2014; Yin et al., 2014). These predictors can be considered generic proxies of resources available to species (Pettorelli et al., 2005) as well as characterizing differences in land surface conditions on a continuous scale (Rocchini et al., 2010; Jung et al., 2019; Randin et al., 2020). It should be noted that the aim of this work is not to identify best possible predictors of local biodiversity, but rather to evaluate most commonly used ones for their predictability and transferability. Other environmental predictors quantifiable from remote sensing exist can provide a better characterization of processes related to the water cycle (e.g., NDWI (Gao, 1996) or energy balance such as LST (Albright et al., 2011)). A more detailed discussion on the potential of remote sensing in predicting local biodiversity is provided in the discussion.

For each site in the PREDICTS databases, I calculated two different remotely sensed predictors that reflect environmental heterogeneity. First, 16-day time series of atmospherically corrected spectral observations (MCD43A v006, (Schaaf et al., 2002)) from the Moderate Resolution Imaging Spectroradiometer (MODIS) sensor on board the Terra and Aqua satellites. These data products are available at a 500 m spatial resolution globally and were downloaded and extracted for each PREDICTS site from Google Earth Engine (Gorelick et al., 2017). Time series of remotely sensed spectral observations often have data gaps caused by clouds or sensor errors. To reduce the number of data gaps, I first aggregated (arithmetic mean) the obtained time series to monthly estimates for each spectral observation (band 1 to 7). The overall proportion of missing data in the aggregated time series was low (mean: 5.9% ± 10.5 SD), nevertheless I subjected the aggregated time series to a missing value imputation using a Kalman smoother on the whole time series (Hyndman & Khandakar, 2008) as implemented in the ‘imputeTS’ R package (Moritz & Bartz-Beielstein, 2017). Whenever the imputation did not converge, a linear interpolation was used to impute missing observations among years. Only data gaps smaller than 5 months were filled in that manner and sites with six or more missing months were excluded from subsequent analyses. From the full time series, I then selected for each site the first year (12 months) of data preceding biodiversity sampling as representation of environmental heterogeneity (Jung et al., 2019).

Second, I calculated from the remaining time series of spectral observations, as proxy of overall photosynthetic activity, the arithmetic mean of the two-band Enhanced Vegetation Index (EVI, Jiang et al., 2008). Photosynthetic activity approximates the condition, structure and availability of plant biomass. Variations in photosynthetic activity have previously been shown to reflect continuous gradients in land cover (Huete et al., 2002; Radeloff et al., 2019) and directly influence local biodiversity measures and life history (Pettorelli et al., 2005; He, Zhang & Zhang, 2009; Oldeland et al., 2010; Jung et al., 2019; Jung, Rowhani & Scharlemann, 2019). Furthermore, I also calculated a measure of overall spectral variability from the satellite sensor data (Rocchini et al., 2010; Rocchini, Hernández-Stefanoni & He, 2015; Randin et al., 2020). Spectral variability is expected to give a more nuanced view on land surface conditions than any single vegetation index, given that it utilizes not two but all spectral bands of the satellite (Rocchini et al., 2010) and thus being able to capture variation among most of spectral scales covered by the MODIS sensor. To capture spectral variability, I first calculated a principal component analysis of all spectral observations (bands 1–7) and then calculated from the first two axes, which on average explained 93% ± 5.92 SD of all variation, the centroid of the resulting bivariate scatter plot. Spectral variability per site was then summarized as the mean Euclidean distance to this centroid. Both environmental predictors, photosynthetic activity and spectral variability are only weakly correlated (Pearson’s r = −0.21, Fig. S1). In total, 21,821 sites had suitable remote sensing data for subsequent analyses, with the remainder (4,028 sites) being sampled either too long ago for sufficient remote sensing coverage from MODIS (2,000 onwards) or having too many data gaps.

Statistical analysis

In the context of this work, ‘predictability’ is defined as the ability to accurately infer a biodiversity measure yij based on the environmental covariates xij among the sites i of a PREDICTS study j (Fig. 1B), and ‘transferability’ as the ability to accurately predict yi based on the environmental covariates xi across studies of the same sampling methodology and taxonomic group (Fig. 1C).

In both predictability and transferability variants prediction accuracy is assessed by calculating for each study the symmetric mean absolute percentage error sMAPEj = 100n∑i=1I⁡|ypredicted−yobserved(|yobserved|+|ypredicted|)| between the observed biodiversity measures ( yobserved) and the ones predicted by the model ( ypredicted) for a given site i. The sMAPE quantifies the percentage error in a model prediction and is bounded between 0% and 100%. Alternative metrics to quantify prediction precision and accuracy exists, however in this case the sMAPE is preferrable for PREDICTS style data owing to its simplicity and inter-comparability between studies that use biodiversity measures of different units and value ranges.

I constructed separate models for each study j and biodiversity measure y in site i, by assuming that yi=αi+βixi+ϵ, where α is the study specific intercept, β a slope coefficient, x the environmental predictor and ϵ an error term. Models of Si were assumed to have Poisson distributed errors and a log-link function ( log⁡y), while models of Ai, PIEi and SIMi−inwere assumed to have Gaussian distributed errors. Pairwise similarities in species composition (Sorensen Index) were related to differences in environmental predictors x in addition to pairwise distance between sites, calculated as log10(x+0.05km) from great circle distances between sites. Here I calculated pairwise absolute difference in mean photosynthetic activity or between spectral centroids of each site (see environmental predictors). For each constructed full model I furthermore calculate an R2 measure as indication of overall variance explained.

To evaluate the predictability and transferability of local biodiversity environment relationships, I constructed in total ten permutation sets, in each of which sites were split into testing (33%) and training (66%) datasets. For evaluating predictability, I removed one third of sites (33%) at random (Fig. 1B), but weighted them by the mean distance to the study centroid, therefore placing extra weight on sites that are less likely to be in close proximity (Roberts et al., 2017). For transferability, instead of individual sites within the same study, I instead sampled and removed 33% for each set sites at random from a different study of comparable methodology in the PREDICTS database (Fig. 1C, methods above). Thus predictability permutations contained samples of training and testing sites for the same study, while transferability permutations contained training sites from one study and withheld testing sites from another comparable one. For studies with sufficient number of sites (>4) I furthermore tested whether considering non-linear relationships (estimated using generalized additive models) resulted in prediction with lower sMAPE, however found that linear models throughout resulted in predictions with smaller errors. However across all ten permutation sets, I iteratively weighted (0–1) this sampling by whether a given study has been sampled before, therefore ensuring that each study is part of both testing and training dataset at least once.

For each respective permutation set, predictability and transferability was then evaluated by using the remaining training data to estimate the regression specified above for each study or group of comparable methodology. I excluded combinations of taxonomic groups, sampling method and sampling unit for which fewer than two studies where available. In total 77.3% of all studies had a matching study of comparable methodology and unit for the same taxonomic group. A table with all recategorized combinations (43) can be found in the Supplemental Materials (Table S1). Using the fitted models I predicted y for the excluded ‘hold-out’ 33% sites and then calculated the average sMAPE for each study in the permutation sets.

Lastly, I explored possible correlates of why sMAPE for some studies is larger than for others for each of the four considered biodiversity measures. I considered a series of variables commonly related to differences in sampling design, species and individual detectability and errors in remotely-sensed environmental predictors. Specifically, I calculated for each study in the permutation sets, the median sampling extent (m) as measure of sample grain, the median sampling duration (days) of the study, the number of sites with a study as measure of effort for the whole study, the average number of samples across sites as effort for area-based sampling effort or the average time sampled (hours) for time-based sampling effort, average accessibility to sites in the study (distance to nearest city in meters) from Weiss et al. (2018), and finally factors related to possible errors in remotely-sensed environmental variables, including the amount of missing data (before gap filling) and the average topographic ruggedness per study using data from Amatulli et al. (2018). To make comparisons across these different units and scales, I standardized all variables before model fitting by subtracting the mean and dividing by one standard deviation.

I fitted linear models allowing partial pooling among studies j (Harrison et al., 2018) by adding a random intercept αk in addition to the overall intercept, e.g., SMAPEj=α+αk+βjxj+ϵ. These kind of models can borrow strength among studies by shrinking individual estimates towards an overall population-wide average (Purvis et al., 2018; Harrison et al., 2018). As random intercept k I used the methodology specific grouping (see methods and Table S1), thus pooling possible correlates among studies of similar methodology. I fitted all possible combinations between the above mentioned variables, including an interaction between sampling extent and sampling effort, finally constructing an average ensemble model of the 5% best performing models. Models were fitted in lme4 (Bates et al., 2015) using the ‘MuMIn’ package in R for model averaging (Bartoń, 2015).

Results

The explanatory power of environmental predictors–photosynthetic activity and spectral variability–in explaining differences in biodiversity varied across biodiversity measures and individual studies. Models fitted with photosynthetic activity explained on average slightly more variance than models fitted with spectral variability, the former having an average R2 of 0.21 (±0.285 SD) compared to an average R2 of 0.19 (±0.284 SD) in the latter. There was considerable variation of R2 values across studies and biodiversity measures (Fig. 2), with species richness on average being best explained by photosynthetic activity (R2 = 0.246 ± 0.311 SD) or spectral variability (R2 = 0.22 ± 0.306 SD). Notably, correlations with species abundance were particularly low, with the R2 being close to 0 (R2 < 0.001) for more than a quarter of all studies (Fig. 2). Meanwhile the difference in explained variance between models using photosynthetic activity compared to spectral variability was lowest for differences in assemble composition (Pearson’s R = 0.922). There were no obvious spatial (Fig. S2) or directional patterns (Fig. S3) in the average explained variance, although some studies notably had high explanatory power regardless of the considered biodiversity measure (Fig. S2).

Figure 2 Explained variance (R²) calculated from models fitted between different biodiversity measures and either photosynthetic activity or spectral variability.

Each point is an individual study in the PREDICTS database with point size indicating the number of sites per study and the colour being a visual indication of density in the plot. A map of the average R2 per study and biodiversity measure can be found in Fig. S2.

When applying local biodiversity models to known (‘Predictability’) or different (‘Transferability’) contexts, the main issue is how accurately such models can predict local biodiversity measures in unknown situations based on the covariates of interest (Fig. 3). Regardless of whether remotely-sensed photosynthetic activity or spectral variability was used as covariate, linear models were reasonably accurate for known contexts in inferring species richness (sMAPE of 19.1%), abundance (11.8%) and evenness (10.3%), but less so when inferring differences in species assemblages (49.3%). Errors in predicting local biodiversity to different contexts were expectedly larger (Fig. 3), whereas particular species richness could be extrapolated relatively poorly (relative error 43.3%) similarly to differences in species assemblages (67.9%), compared to abundance (25.4%) or evenness (14.3%). Notably, when local biodiversity models are used to extrapolate richness to different contexts, the sMAPE was larger than 50% in 35% of all studies, compared to 8.1% and 4.7% for abundance and evenness (Fig. 3).

Figure 3 Distribution of the symmetric mean absolute percentage error (sMAPE) of biodiversity measures calculated from models using photosynthetic activity or spectral variability.

Larger values (range 0 to 100) indicate a larger prediction error. Colours differentiate between models that evaluate Predictability and Transferability (see Methods). Point error ranges show the arithmetic mean and standard deviation of the sMAPE.

There were also considerable differences in prediction error, as quantified by the sMAPE, among taxonomic groups. Across taxonomic groups and biodiversity measures the sMAPE was larger when predictions were extrapolated to novel contexts compared to predictability, particularly so for reptiles (ΔsMAPE = 21.3%) and mammals (ΔsMAPE = 20.8%), with the greatest difference being for reptile species richness (ΔsMAPE = 33%) and abundance (ΔsMAPE = 28%). The transferability of fungi (sMAPE = 7.5%), and bird (sMAPE = 9.1%) assemblage evenness was overall the lowest, while predictability was best for evenness and abundance of fungi (sMAPE = 5.11%) and plants (sMAPE = 9.65%). Fungi and Plants had across biodiversity measures the lowest sMAPE in predictability and transferability (Fig. 4). Overall, assemblage composition of vertebrates was the most poorly predicted with sMAPE estimates well over 50% throughout (Fig. 4).

Figure 4 Average error (sMAPE) across models for predictability and transferability.

Errors were averaged (lines indicating standard deviation) across models with different biodiversity measures (shapes) and taxonomic group (colours). Shown only for models using photosynthetic activity as predictor as spectral variability results were broadly comparable in overall patterns (Fig. S4).

I also explored across studies which factors helped explain differences in prediction error, as quantified by the sMAPE (Fig. 5). Across biodiversity measures, having a greater number of samples per site most effectively reduced the sMAPE (Δβ = −3.14) for transferability, and so did sample duration but to a lesser degree (Δβ = −0.98). Meanwhile a greater number of sites per study on average increased the sMAPE (Δβ = 2.23). Patterns of comparison results were broadly similar between transferability (Fig. 5) and predictability (Fig. S5), although notably a study being more accessible resulted in an average larger reduction in the sMAPE (Δβ = −1.02) for predictability (Fig. S5). Overall variance explained by these factors in the average model was relatively low ( Rmarginal2 = 0.08, Rconditional2=0.14).

Figure 5 Averaged and standardized model coefficients of variables that best explain differences in sMAPE.

Standardized coefficients smaller than zero indicate that increases in a given variable reduce study-specific prediction errors, while coefficients greater than zero increase the error. Shapes distinguish different biodiversity measures (as in Fig. 3). Standardized coefficients shown for transferability permutations only as predictability results follow similar patterns (Fig. S5).

Discussion

In this work I make use of a large database of local biodiversity survey records and remote sensing data to evaluate the predictability and transferability of biodiversity-environment relationships, e.g., the ability of models to infer local biodiversity measures in known and novel contexts. Particular emphasis is placed on differences among biodiversity measures, taxonomic groups and sampling circumstances. I found that the explanatory power of biodiversity-environment was relatively low for most studies (Fig. 2). This aligns with a previous meta-analysis that found that relationships between biodiversity measures and photosynthetic activity cannot always be established (Duncan et al., 2015). I also discovered that prediction errors are on average lowest for evenness and abundance, and, maybe unsurprisingly, generally larger when models predictions are transferred to novel contexts (Fig. 3). Biodiversity measures of sessile organisms were on average more precisely predicted (Fig. 4), although not by much with predictions errors generally larger than 25% compared to observed values, particularly so for differences in species assemblage composition. Overall these results shed some doubts on the predictability and transferability of biodiversity measures, although they have to be interpreted in the context of the individual studies (Fig. 5) and ultimately in what is an acceptable accuracy to achieve with such predictions.

Indeed, it is not formerly defined what makes a prediction better or worse based on quantitative measures such as the cross-validated error metrics used in this study. According to Yates et al. (2018) ‘transferability’ is broadly defined as the capacity of a model to produce predictions for a new set of predictor values that differ from those on which the model was trained. Similarly predictability can be understood as the capacity of a model to infer held-out observations (Fig. 1). In this context, a good precision could be understood as a model that demonstrates transferability errors smaller or comparable to errors inherent in model inferences or that don’t exceed an apriori set threshold. I found that the predictability of local biodiversity measures was overall reasonable good with errors being smaller than 25% in most cases (Fig. 3), although particularly differences in assemblage composition were poorly predicted. This might indicate that photosynthetic activity and spectral variability are useful predictors for quantifying differences in local biodiversity measures, although the variance explained varied considerably across studies (Fig. 2). In contrast I found that errors associated with transferability of biodiversity measures can be considerable, exceeding 50% relative to the original measure for species richness and differences in assemblage composition in many studies (Fig. 3). This is especially relevant, since a number of studies spatially extrapolated local biodiversity estimates, e.g., species richness or abundance, to unsampled areas based on environmental predictors (König, Weigelt & Kreft, 2017; Phillips et al., 2019; van den Hoogen et al., 2019). These approaches assume that local biodiversity-environment relationships are transferable to new, unsampled environments and the results by this work indicate that this often entails considerable errors. Ideally models are evaluated on their ability to accurately reproduce their data in novel contexts (Jung et al., 2017), quantify the uncertainty in doing so, or alternatively limit predictions to areas within the models applicability (Mesgaran, Cousens & Webber, 2014; Meyer & Pebesma, 2021), and the results of this work further highlight that prediction errors should be reported.

Biodiversity measures for certain taxonomic groups might be easier to predict than others owing to the dynamics, drivers and mechanisms underlying them (Magurran, 2004). Indeed previous studies have found species abundance to be stronger correlated with photosynthetic activity than other measures (Oldeland et al., 2010; Duncan et al., 2015). Similarly, I found that abundance-based biodiversity measures–e.g., abundance and evenness–had overall lowest precision errors (Fig. 3). A potential mechanism could be that a greater photosynthetic activity or spectral variability is indicative of resources available to species populations, facilitating population growth (Hurlbert, 2004; Pettorelli et al., 2006). While species richness had the largest average explained variance compared to other biodiversity measures, it performed considerably poorer when evaluated in predictions (Fig. 3). Possibly, the processes underlying patterns of local species richness, such as colonization and extinction, might cause simple predictions to fail (Chase, 2003), unless the spatial-temporal dynamics of environmental predictors are taken into account (Fernández, Román & Delibes, 2016). Similarly, the fact that both predictability and transferability errors were on average lowest for more sessile organisms such as Fungi and Plants (Fig. 4), likely indicates that similar important processes mediate biodiversity-environment relationships. Overall this study highlights the benefit of comparing relationships across a range of studies and biodiversity measures (Stein, Gerstner & Kreft, 2014; Duncan et al., 2015), revealing that biodiversity-environment relationships are not universally strong.

Investigating as to what factors best explain prediction errors can help to improve future monitoring and modelling efforts. Among the most important factors that resulted in overall smaller prediction errors was the average number of samples per sites (Fig. 5), which can be considered a simplified metric of sampling completeness. Given that errors were smaller for sites with many samples, it could be that many species communities in the PREDICTS database have not been comprehensively sampled, if one assumes that biodiversity-environment relationships are strongest in equilibrium. There are ways to account for detectability and observation biases (Royle, Nichols & Kéry, 2005), which however was not feasible for the studies in the PREDICTS database given the heterogeneity of sampling information. Thus better standards for sampling techniques and monitoring are advisable to enable better comparability (Montgomery et al., 2021).

Interestingly, and in contrast to previous studies (Chase & Knight, 2013), differences in sample grain, e.g., the linear scale of sampling, did not help to explain why biodiversity measures could be better predicted in some studies. A likely explanation is that the contrasts between sampling extents are relatively small (most studies in the PREDICTS database were sampled at scales between ~1 and 4,000 m). Scale-dependent effects might only become apparent at spatial scales that go beyond the local scale. A spatial mismatch at the lower end, e.g., that the grain of the used MODIS data is too coarse to be matched to the extent of sampling in PREDICTS studies, could be another explanation, however previous studies that used very-high resolution satellite imagery (<10 m) did not find much more accurate predictions than presented here (Dalmayne et al., 2013; Hofmann et al., 2017). Other, non-explored factors could further explain differences in prediction error, such as for instance preceding changes in environmental predictors (Jung et al., 2019; Jung, Rowhani & Scharlemann, 2019) or a better accounting of differences in species traits (Duncan et al., 2015; Regos et al., 2019). Future efforts could evaluate if inter- and intra-specific variability of species traits can be more precisely linked to differences in environmental heterogeneity.

In this work I used photosynthetic activity and spectral variability as measures of environmental heterogeneity, acknowledging that other characterizations of environmental heterogeneity (e.g., soil, micro-climate) could be more important (Stein & Kreft, 2015). The rationale behind focusing on photosynthetic activity and spectral variability in this work was the assumption that they can serve as general broad predictive measures across taxa and regions. There are, however, also a number of other remote-sensing measures, such as for example land-surface temperature or remotely-sensed moisture (Albright et al., 2011; Regos et al., 2022), as well as different summary statistics of such measures (maximum, minimum, variation), which for some local contexts might capture important dynamics and lead to an improved prediction with lower errors. The finding that prediction errors were lowest for plants and fungi could be related to the fact that photosynthetic activity is more closer related to the abundance of these taxa, than for other taxonomic groups, where only indirect correlations (resources for herbivores, differences in land cover) could be the most likely explanation and local factors might have a deciding influence on explaining these differences (Jung et al., 2017). Yet, focussing solely on remotely-sensed variables ensures global consistency and is frequently used to predict local biodiversity measures (Dalmayne et al., 2013; Hobi et al., 2017; Hofmann et al., 2017; Randin et al., 2020).

Another key limitation is that environmental heterogeneity is not necessarily related to differences in land use and land-use intensity, for which the PREDICTS database was explicitly designed (Purvis et al., 2018). Indeed, it could be that the potential of remotely sensed environmental heterogeneity in predicting local biodiversity measures has been exaggerated, and better and more direct characterizations of land use and its management from remote sensing have to be developed. Previous studies using PREDICTS data found that even using the local land-use categorizations explained few differences in biodiversity measures, with most variance being ‘soaked up’ by study-specific differences (Newbold et al., 2015, 2018; De Palma et al., 2015). Further, given the complexities of local species community assembly (Chase, 2003; Leibold et al., 2004), any claim of a direct prediction of ‘biodiversity’ through remotely-sensed proxies should thus be interpreted with caution and only under consideration of prediction uncertainties. Remote sensing measures are at best able to capture changes in habitat extent or condition; and those changes do not necessarily correlate strongly with changes in biodiversity measures. Future work should ideally focus on the principal mechanisms of species community assembly, their practical incorporation into models and how remote sensing can assist in capturing relevant predictors.

Conclusions

The findings presented in this study have particular implications for spatial projections of local biodiversity-environment relationships. Ecological models can and should be used for predictions (Houlahan et al., 2017; Tredennick et al., 2021); however, caveats and limitations should be better identified, communicated and hopefully build upon. We need to create models that enable biodiversity-environment relationships to be more predictable across scales and novel contexts, especially when applied to conservation contexts (Santini et al., 2021). Remote sensing data can be used for global consistent characterizations of environmental heterogeneity, but given the considerable drops in precision for transferability, this work recommends that prediction errors in projecting local biodiversity measures are clearly communicated and quantified. To improve future biodiversity predictions I further propose that models (a) should be evaluated comprehensively based on their ability to create accurate predictions, (b) account better for underlying hierarchies and sampling effects, (c) ensure that environmental predictors are quantified in a globally replicable and transparent way. Quantitative correlative models might not be the most precise in many situations, but that does not invalidate their use if shortcomings are appropriately communicated.

Supplemental Information

Supplemental Information 1 Average photosynthetic activity visualized in relation to spectral heterogeneity of each site.

Sites coloured by Biome according to Dinerstein et al. (2017).

Click here for additional data file.

Supplemental Information 2 Map of average explained variance (R²) of models relating biodiversity measures against environmental predictors.

Each dot represents the centre coordinates of a study with size and colour indicating the explained variance.

Click here for additional data file.

Supplemental Information 3 Distribution of predictability regression coefficients fitted per study shown for each biodiversity measure (horizontal) and environmental predictor (vertical).

Error bars show the mean and 1 standard deviation of the regression coefficients.

Click here for additional data file.

Supplemental Information 4 Average error (sMAPE) across models for predictability and transferability and using spectral variability as predictor.

Colours and shapes as in Figure 4.

Click here for additional data file.

Supplemental Information 5 Averaged and standardized model coefficients of variables that best explain differences in sMAPE for predictability.

Click here for additional data file.

Supplemental Information 6 Distribution of attributes of the subset of PREDICTS sites and studies used in this work.

Shows the (a) log-transformed number of sites per single study, (b) log-transformed number of samples per site in a study, (c) log-transformed average distance between sites within a study and (d) log-transformed sampling period (start to end) in days.

Click here for additional data file.

Supplemental Information 7 Combination of sampling methodology pairing used.

Taxonomic group indicates the broad taxonomic group used for grouping studies, Sampling method groups together the type of surveying method applied in a study and sampling unit indicates whether effort was measured as number of plots, area covered or time sampled. The concatenation of all columns returns a methodology specific grouping that is used to construct permutations and models.

Click here for additional data file.

Additional Information and Declarations

Competing Interests

Author Contributions

Data Availability

The author declares that there are no competing interests.

Martin Jung conceived and designed the experiments, performed the experiments, analyzed the data, prepared figures and/or tables, authored or reviewed drafts of the article, and approved the final draft.

The following information was supplied regarding data availability:

The raw biodiversity data is available at the Natural History Museum:

Lawrence Hudson; Tim Newbold; Sara Contu; Samantha L. L. Hill et al. (2016). The 2016 release of the PREDICTS database [Data set]. Natural History Museum. https://doi.org/10.5519/0066354.

All intermediate analysis results and code are available at GitHub:

https://github.com/Martin-Jung/TransferabilityPREDICTS.

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
