# Peer review of "Predictability and transferability of local biodiversity environment relationships"

_PeerJ, doi:10.7717/peerj.13872_

## Round 0.1 · original submission · Major Revisions

Both reviewers appreciate the work you've put into this analysis, and have offered some suggestions for a revision. In particular, please try to incorporate the additional citations (Reviewer 1) and address the concerns about the experimental design (Reviewer 2).

·

Basic reporting

General comments

Dear author,

I find the work very valuable and ambitious. The manuscript investigates local biodiversity-environment relationships and, in particular, the ability of biodiversity models to accurately predict different biodiversity metrics across a wide range of environmental conditions. The manuscript is very valuable, the methods are sound. However, I have some doubts regarding the tight selection of predictor variables (solely based on spectral diversity metrics and photosynthetic activity) that can limit the scope of the main findings. Also, it is not clear how the author can be measuring model transferability with a cross-validation procedure.

I miss some recent literature on the application of remotely sensed ecosystem functioning variables and their incorporation to ecological modelling, both from the remote sensing and ecological modelling literature (see our recent review paper Regos et al. 2022, and reference therein). Please see other comments bellow.

Specific comments

Abstract

Line 25-26. I am not fully convinced about the way “predictability” and “transferability” are defined in the abstract, it might be confusing for readers. I would suggest defining the “predictive capacity” of a model as the model’s ability to accurately predict local biodiversity measures within (“cross-validation”) and beyond the range of data used for calibration (“transferability”). The generalization of a model outside the range used for calibration allows to extrapolate the model in space and time (spatiotemporal transferability), a validation procedure that often requires independent datasets. Such independent datasets are not commonly at hand so ecological modelers are forced to use cross-validation procedures (split-sample approach, i.e., repeatedly, and randomly leaving out a subset of data used for calibration) to assess model accuracy. This especially relevant when calibrating models to predict biodiversity changes under future scenarios, that never can be validated with independent data. These issues are well described in lines 58-63 but not in the abstract. Please be consistent along the whole text.

Introduction

Lines 40-41. Not sure if resources can be defined as “diversity of habitats and landscapes, or availability and structural complexity of vegetation or rocks”. I see it in the other way around, ‘habitat’ is the place where an organism can live due to the availability of light, nutrients, water resources, etc. The higher the environmental heterogeneity the higher the availability of different resources.

The author justifies the inclusion of remote sensing variables into biodiversity models because these variables provide robust quantification of environmental heterogeneity at high spatial and temporal resolution (see lines 83-97). However, it is not clear to me, what does the author mean by “environmental heterogeneity”? The term is a bit ambiguous. Please be more specific about the potential advantages of using remote sensing variables as predictor variables in biodiversity models, according to recent literature. Among others, remotely sensed variables have an added value as niche descriptors, complementary to those traditionally used, thus providing a more complete view of the species’ ecological niche and habitat characterization in space and time; allowing incorporating intra- and inter-annual variability of habitat dynamics and key ecosystem processes related with different ecosystem functioning dimensions (not only related with carbon cycle but with energy balance). They can be measured systematically at different spatio-temporal scales, facilitating a cross-scale, standardized, repeatable and cost-effective model-assisted monitoring, and long-term data archives such as those provided by Landsat or Terra/Aqua satellites offer the possibility to assess decadal changes at fine resolutions suitable for species modelling applications to conservation. In addition to Randin et al. (2020), please see Regos et al. (2020) and references therein.

Regos A, Gonçalves, J., Arenas‐Castro, S., Alcaraz‐Segura, D., Guisan, A., & Honrado, J. P. (2022). Mainstreaming remotely sensed ecosystem functioning in ecological niche models. Remote Sensing in Ecology and Conservation. https://doi.org/10.1002/rse2.255 .

Lines 160, add “be attributed”

Methods.
Lines 163-170. Although the advantages of using RS variables are better described in methods than in the introduction, I still consider the selected variables a bit biased toward vegetation spectral indices (i.e., carbon cycle). There is a bunch of potential candidate variables that can be derived from remote sensing related with water cycle (e.g., NDWI, soil moisture) and energy balance (LST, albedo) that are not here considered, which can affect the performance and transferability of models based on such environmental-biodiversity relationships.

Lines 193. EVI, as many other vegetation spectral indices, is a good proxy for photosynthetic activity, but the annual mean of a time series has been widely accepted as good descriptors of primary productivity (see these references, among others). I think that the term 'primary productivity' would better characterize this type of predictors.

Pettorelli, N., Schulte to B€uhne, H., Tulloch, A., Dubois, G., Macinnis-Ng, C., Queir?os, A.M. et al. (2018) Satellite remote sensing of ecosystem functions: opportunities, challenges and way forward. Remote Sensing in Ecology and Conservation, 4,71–93. https://doi.org/10.1002/rse2.59
Fernández, N., Román, J. & Delibes, M. (2016) Variability in primary productivity determines metapopulation dynamics. Proceedings of the Royal Society B: Biological Sciences, 283, 20152998. https://doi.org/10.1098/rspb.2015.2998.
Phillips, L.B., Hansen, A.J. & Flather, C.H. (2008) Evaluating the species energy relationship with the newest measures of ecosystem energy: NDVI versus MODIS primary production. Remote Sensing of Environment, 112, 4381–4392. https://doi.org/10.1016/j.rse.2008.08.002
Paruelo, J.M., Burke, I.C. & Lauenroth, W.K. (2001) Land-use impact on ecosystem functioning in eastern Colorado. Global Change Biology, 7, 631–639.

Lines 198-210. I think the author refers to spectral diversity metrics developed by Rocchini et al. But for readers unfamiliar with the approach, it might be not clear what this set of predictors mean from an ecological viewpoint. Please try to be more descriptive with this set of variables and better justify why the predictor variables are only focused on the photosynthetic activity and spectral diversity metrics. I think that the approach is setting aside important descriptors of habitat dynamics (which might explain the overall low R2).

Lines 235-244. I have some doubts regarding the validation procedure. It seems that both “predictability” and transferability” were measured through cross-validation, i.e., splitting the dataset in testing (33%) and training (66%) subsets. I am wondering to what extent these models are being actually extrapolated (rather than interpolated). It is not clear to me if model transferability is really measured. Please clarify it.

Overall, in discussion and conclusions, I would reinforce the importance of including relevant ecological variables that were not included in this study. Remote sensing technologies and data already offer a wider range of potential variables that were not considered. EVI-based variables can work well for forest-dwelling species but metrics such as annual mean or standard deviation of Land Surface Temperature or Albedo can be extremely important to explain the ecological fitness if keystone species at the top of the food chain (see e.g., Regos et al 2021).

Regos, A., Tapia, L., Arenas-Castro, S., Gil-Carrera, A., & Domínguez, J. (2021). Ecosystem Functioning Influences Species Fitness at Upper Trophic Levels. Ecosystems, 1-15. https://doi.org/10.1007/s10021-021-00699-5

Congratulations for this valuable work.

I hope my comments can help improve the manuscript.

Al the best,

Experimental design

The manuscript is very valuable, the methods appropriate. However, I have some doubts regarding the selection of predictor variables (solely based on spectral diversity metrics and photosynthetic activity) that can limit the scope of the main findings. Also, it is not clear how the author can be actually measuring model transferability with a cross-validation procedure. See comments above.

Validity of the findings

The validity of the main findings can be in some way constrained by the type of co-variate used to fit the models. I think that this potential drawkack should be clearly included in the discussion to avoid misslieading conclusions.

Additional comments

'no comment'

Reviewer 2 ·

Basic reporting

In this study it is tested whether satellite-derived indicators of environmental heterogeneity can be used to predict biodiversity with a focus on whether modelled relationships can be transferred to other areas. This is an important topic considering the increasing request for (global) information on biodiversity patterns and the increasing discussion on how these information can be provided based on limited field data and remote sensing.

The manuscript is written in professional English and is well structured. The provided figures are relevant and well designed. Unfortunately the study could not be reproduced because neither code nor raw data were made available.

I have several concerns about the design and the relevance of findings that I will outline below.

Experimental design

The introduction section makes the context clear in most parts but could be improved. I feel it is not sufficiently placed into context and does not clearly highlight the knowledge gap that is to be filled by this study. Also the thread could be improved. E.g. the author introduces soil and climate as predictor (line 66) but this is not tested here. The author also talks about inconsitencies in model assessment (line 64) but this is not approached in this study. So I think that it summary, the introduction could be better tailored to the aims of this study. The third hypothesis “(iii) unexplained variation is predominantly linked to differences in study design, e.g. spatial scale and sampling duration.” is not clearly delineated. It appears more reasonable that most parts of unexplained variation is due to limited information about biodiversity patterns present in the predictors rather than on e.g. spatial scale of the study design.

Aim of the study is to test how well predictive models of biodiversity can be used to make predictions beyond study regions. In this study, only a very limited set of predictors are used and modeling approaches are limited to linear models. The author says that the aim is not to reveal most relevant predictors but to use commonly applied ones (Line 173) but he does not convincingly outline whether these predictors (individually) are used in practice to model biodiversity patterns. I think that conclusions derived by this design cannot be generalized because the findings are likely not transferable to other modelling approaches that usually differ from what is presented here (e.g. environment described by climate, soil, etc rather than only spectral data; more complex modelling approaches commonly using machine learning strategies rather than linear models). A main question in this context is if we can assume that the relationships are linear? The paper does not convince that this modelling strategy is adequate.

My major concerns is the question of scale. The study is based on the idea that spectral diversity is a proxy for environmental heterogeneity which again is a proxy for biodiversity. The hypothesis, however, only makes sense if the resolution of the spectral data allows to reflect the relevant heterogeneity. With a 250m MODIS pixel size, it reflects large-scale heterogeneity but no small-scale variability which however might be relevant for some taxonomic groups. Therefore, I miss a convincing explanation why MODIS is a reasonable choice. Can it be convincingly explained that large scale heterogeneity is a relevant proxy for biodiversity and is this in line with other studies (keeping in mind that this paper is not aiming at suggesting new proxies but rather to test the transferability of well established approaches for biodiversity prediction). Also, do plots of a PREDICT study covered all land cover types of an area or where they limited to single land cover types. If so, the design is problematic because we would have a high spectral diversity if we had several land cover types. But is this a relevant proxy for biodiversity e.g. within forest sites?

Besides, some details on the data are missing. How many sites, how many plots per site, how are sites and plots distributed etc. Maybe some exemplary figures could be helpful here? Also the predictors are not clear. E.g. what is the source for the predictors on photosynthetic activity? Line 165 suggests that different resolutions are considered but I don’t find the description relating to this.

Validity of the findings

Following my concerns on the experimental design, I find the results and discussion not entirely convincing. The results indicate it’s hard to predict biodiversity and even harder to transfer the models, but we don’t learn why. Although Figure 5 indicates factors that influence the prediction performance, the general performance remains low and I suspect that this is rather due to the limited modelling approach which considers only limited predictor variables and simple models.

The conclusions need a lot of improvement. They are not fully supported by the data and the key findings with regard to the hypotheses are not made clear. E.g. the recommendation of applying the “area of applicability” of the model is not analyzed. Also to “ensure that environmental predictors are quantified in a globally replicable and transparent way” is a conclusion that can not really be derived from the results.

Additional comments

Minor
Line 20: “However so far little work has been done on quantitatively evaluating if
and how accurately local biodiversity measures can be predicted.”
I disagree. There are plenty of studies where local observations are used to make predictions for larger areas and all of them assess how accurate predictions can be made. There might be flaws in the way this is done, but I don’t think the sentece that little work has been done on this is correct. Similarly Line 33-34 states “that future predictions should be evaluated based on their accuracy and inherent uncertainty, and ecological theories be tested against whether we are able to make accurate predictions from local biodiversity data.” Again, usually they are evaluated based on this.

Line 330 claims that the paper “comprehensively evaluated the predictability and transferability of biodiversity-environment relationships.” Since the environment is only described by a limited set of predictors and modelling approaches are limited too, I find the word “comprehensive” not appropriate here.

Line 413: “Spectral availability” is not the correct wording

Fig 1: The different bars in the right part of the figures are not clear to me

Fig 2 I think showing N and density at the same time is too much because where the density is high, N cannot be interpreted anymore. This is only possible for the outliers.

---

## Round 0.2 · accepted · Accept

The manuscript has been improved and is more clear; thank you for addressing the reviewers' comments.